# Beyond emotions: Social cognitive predictors of COVID-19 vaccination intentions before and after vaccine roll-out

Athina Manoli[1,2‡], Maria Kyprianidou[1,3,4‡], Demetris Lamnisos[5], Jelena Lubenko[6], Giovambattista Presti[7], Valeria Squatrito[7], Marios Constantinou[8], Christiana Nicolaou[1], Savvas Papacostas[9], Gökçen Aydın[10], Yuen Yu Chong[11], Wai Tong Chien[11], Ho Yu Cheng[11], Francisco Ruiz[12], Maria Belen Garcia-Martin[13], Diana P. Obando-Posada[14], Miguel Segura-Vargas[12], Vasilis S. Vasiliou[15], Louise McHugh[16], Stefan Höfer[17], Adriana Baban[18], David Dias Neto[19], Ana Nunes Da Silva[20], Jean-Louis Monestès[21], Javier Alvarez-Galvez[22,23], Marisa Paez-Blarrina[24], Francisco Montesinos[25], Sonsoles Valdivia-Salas[26], Dorottya Ori[27,28], Bartosz Kleszcz[29], Raimo Lappalainen[30], Iva Ivanović[31,32], David Gosar[33,34], Frederick Dionne[35], Rhonda Merwin[36], Maria Karekla[3‡], Andrew Gloster[37‡], Angelos Kassianos[1‡*]

1 Department of Nursing, Faculty of Health Sciences, Cyprus University of Technology, Limassol, Cyprus, 2 Centre for Psychiatry & Mental Health, Wolfson Institute of Population Health, Barts & The London School of Medicine & Dentistry, Queen Mary, University of London, United Kingdom, 3 Department of Psychology, University of Cyprus, Nicosia, Cyprus, 4 Department of Social and Political Sciences, University of Cyprus, Nicosia, Cyprus, 5 Department of Health Sciences, School of Sciences, European University Cyprus, Nicosia, Cyprus, 6 Psychological Laboratory, Faculty of Public Health and Social Welfare, Riga Stradins University, Riga, Latvia, 7 Kore University Behavioral Lab (KUBeLab), Department of Human and Social Sciences, University of Enna "Kore", Enna, Italy, 8 Department of Social Sciences, School of Humanities and Social Sciences, University of Nicosia, Nicosia, Cyprus, 9 Cyprus Institute of Neurology and Genetics, Nicosia, Cyprus, 10 Department of Guidance and Psychological Counseling, TED University, Ankara, Türkiye, 11 The Nethersole School of Nursing, Faculty of Medicine, The Chinese University of Hong Kong, Hong Kong SAR, China, 12 Department of Psychology, Fundación Universitaria Konrad Lorenz, Bogotá, Colombia, 13 Department of Psychology and Education, Universidad de Loyola, Sevilla, Spain, 14 Department of Psychology, University of La Sabana, Chía, Colombia, 15 Department of Psychology, Royal Holloway, University of London, United Kingdom, 16 School of Psychology, University College Dublin, Dublin, Ireland, 17 Department of Psychiatry II, Medical University Innsbruck, Innsbruck, Austria, 18 Department of Psychology, Babes-Bolyai University, Cluj-Napoca, Romania, 19 APPsyCI-Applied Psychology Research Center Capabilities & Inclusion, ISPA-Instituto Universitário, Lisbon, Portugal, 20 CICPSI, Faculdade de Psicologia, Universidade de Lisboa, Lisbon, Portugal, 21 LIP/PC2S, Université Grenoble Alpes, Grenoble, France, 22 CS2 DataLab, University Research Institute for Sustainable Social Development, University of Cádiz, Jerez, Spain, 23 Department of General Economy (Sociology area), Faculty of Health Sciences, University of Cádiz, Cádiz, Spain, 24 Instituto ACT, Madrid, Spain, 25 Department of Psychology, Faculty of Biomedical and Health Sciences, Universidad Europea de Madrid, Madrid, Spain, 26 Department of Psychology and Sociology, Universidad de Zaragoza, Zaragoza, Spain, 27 Institute of Behavioural Sciences, Semmelweis University, Budapest, Hungary, 28 Department of Mental Health, Heim Pal National Pediatric Institute, Budapest, Hungary, 29 Behawioralnie, Poland, 30 Department of Psychology, University of Jyväskylä, Jyväskylä, Finland, 31 Department of Child Psychiatry, Institute for Children's Diseases, Clinical Centre of Montenegro, Podgorica, Montenegro, 32 Centre for early development, Clinical Centre of Montenegro, Podgorica, Montenegro, 33 Department of Child, Adolescent and Developmental Neurology, Children's University Hospital, University Medical Centre Ljubljana, Ljubljana, Slovenia, 34 Department of Psychology, Univerisity of Ljubljana, Ljubljana, Slovenia, 35 Département de Psychologie, Université du Québec à Trois-Rivières, Trois-Rivières, Canada, 36 Department of Psychiatry and Behavioral Science, Duke University, Durham, United States of America, 37 Division of Clinical Psychology, Faculty of Behavioural Sciences and Psychology, University of Lucerne, Lucerne, Switzerland

‡ These authors share first and last authorship on this work.
* angelos.kassianos@cut.ac.cy

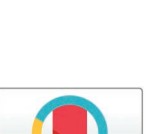

**Data availability statement:** The anonymised dataset and analysis code supporting the findings of this study are openly available on the Open Science Framework (OSF) at: https://osf.io/pr3se/overview.

**Funding:** The authors received no specific funding for this work.

**Competing interests:** The authors have declared that no competing interests exist.

## Abstract

Understanding the drivers of COVID-19 vaccination intentions remains relevant as public health systems prepare for future pandemics. This study examined how emotional and social-cognitive factors influence COVID-19 vaccination intentions during two key phases of the COVID-19 pandemic: before (April-June 2020) and after (January-February 2021) vaccination rollout. A total of 586 adults completed an online survey assessing beliefs about COVID-19, self-efficacy to adhere to protective behaviours, perceived stress, affect, psychological flexibility, and prosociality. Self-efficacy, prosociality, psychological flexibility and positive affect significantly declined after vaccination rollout. Higher self-efficacy and perceived severity of the disease consistently predicted stronger vaccination intentions across time points. Perceived susceptibility was negatively associated with vaccination intention before, but not after rollout. The psychological variables were not significant predictors of intentions. These findings underscore the importance of social-cognitive factors, especially self-efficacy and perceived severity, in shaping vaccination-related decisions, with implications for designing effective communication strategies in future health emergencies.

### Introduction

The COVID-19 pandemic has brought the importance of vaccination to the forefront of public health. Individuals' intentions to get vaccinated changed throughout the pandemic [1], with willingness to vaccinate generally increasing after the first COVID-19 vaccinations became available [2–6]. Understanding the factors that influence vaccination intentions is essential for designing effective public health strategies, particularly in preparation for future infectious disease outbreaks.

A growing body of research has identified multiple psychological and social-cognitive factors that influence vaccination intentions, including perceptions of risk, trust in health institutions, and confidence in one's ability to follow recommended behaviours [7–12]. However, existing findings are drawn from single-country, cross-sectional studies, which may limit the generalisability of findings [13,14]. Longitudinal studies with participants across countries are therefore needed to capture within-person variation in vaccination intentions and their underlying psychological drivers.

The Health Belief Model [15] provides a useful framework for understanding vaccination-related decision making. The model suggests that an individual's health-related behaviours can be explained by the perceived threat to illness or disease (perceived susceptibility) and the beliefs about the consequences of the disease (perceived severity). In the context of COVID-19, studies have shown that perceived susceptibility and severity predicted lower willingness to vaccinate [16–18]. For instance, a cross-sectional study in the United States found that decreased perceived susceptibility, and severity related to lower willingness to vaccinate [19], while data from Malaysia showed that greater perceived benefits of COVID-19

vaccinations, and greater perceived susceptibility were positively associated with intentions to take the COVID-19 vaccinations when they would be available [16].

Psychological traits also play a role in vaccination intentions. For example, greater psychological flexibility (i.e., the ability to adapt to changing circumstances and be open to engaging in new experiences) has been linked to stronger vaccination willingness in several studies [20,21], while higher self-efficacy, or confidence in one's ability to perform protective behaviour, consistently predicts positive vaccination intentions [22]. Evidence further suggests that self-efficacy may mediate the relationship between psychological flexibility and vaccination hesitancy, suggesting that more flexible individuals may feel more capable of making informed health decision [23].

Sociodemographic and affective factors are also relevant to vaccination intentions. Hesitancy tends to be higher among younger individuals, women, those with lower income or education levels, or who live alone or in small communities [24–27]. Emotional and psychological characteristics – including perceived stress, anxiety and depressive symptoms – have been linked to lower vaccination intentions, whereas positive affect and confident attitudes tend to predict greater willingness to vaccinate [9,28,29]. Together, these findings highlight the multifaceted nature of vaccination decision-making and set the context for examining how much influences may evolve.

Despite this growing evidence, most research has relied on cross-sectional data collected early in the pandemic and prior to vaccination rollouts [25–27,29,30], offering limited understanding of within-person changes in vaccination intentions and their psychological determinants over time. Although several longitudinal studies have traced shifts in public willingness to vaccinate following vaccination rollouts [1,4,6,31], these largely capture population-level trends, rather than dynamic evolution of individuals' belief, efficacy perceptions, and motivations. Recent population-level analyses indicate that overall physical and mental health showed non-linear patterns of disruption and partial recovery across the pandemic period, reflecting the broader societal and behavioural adjustments that accompanied evolving public health measures [32]. However, evidence from direct comparisons of the psychological factors affecting vaccination intentions before and after vaccination rollouts within the same individuals remains scarce. Understanding these within person changes can inform more effective strategies to strengthen vaccination efforts, enhance public trust, and guide preparedness for future epidemics or pandemics.

The current study is part of a large multinational prospective study, including data from 33 countries. We examined how social -cognitive factors that are included in behavioural models such as HBM (perceived severity, perceived susceptibility, self-efficacy) and psychosocial/emotional variables (perceived stress, psychological flexibility, prosociality, positive and negative affect) predicted COVID-19 vaccination intentions, comparing responses from before (April-June) and after (January -February) the initial vaccination rollout. Based on previous research, we hypothesised that both social-cognitive and emotional factors would be associated with vaccination intentions, with social-cognitive factors expected to have stronger effects. Understanding these mechanisms can guide more effective and targeted public health strategies to improve vaccination uptake- particularly as the world prepares for future pandemics and seeks to build durable population immunity [13,33].

## Materials and methods

### Ethics statement

The study was approved by the Cyprus National Bioethics Committee (ЕП 2020.01.60). All participants provided informed consent electronically prior to participation.

### Participants and procedure

The study sample included individuals aged 18 years or older who were able to read one of the 18 languages: Chinese, Dutch, English, Finnish, French, German, Greek, Hungarian, Italian, Latvian, Montenegrin, Persian, Polish, Portuguese, Romanian, Slovenian, Spanish, and Turkish.

Data were collected using an online survey administered via RedCap (https://redcap.ucy.ac.cy/). The survey was distributed through university mailing lists (students and staff), institutional websites, social media platforms (e.g., Facebook,

Twitter), local media (e.g., newspapers, radio), professional networks, hospitals and health centres, and community organisations (e.g., churches, schools). The recruitment strategy used was designed to recruit a socio-demographically diverse sample.

The first phase of data collection occurred between 07 April and 07 June 2020. The second phase was conducted between 1 January and 10 February 2021, and invitations were sent to participants who had completed the first survey. During both periods, most participating countries had active COVID-19 restrictions in place. At Time 1, COVID-19 vaccinations were not yet available in any participating countries; by Time 2, vaccination rollouts had begun in nearly all countries represented in the sample.

## Measures

All instruments used in the study were widely used and psychometrically valid. Measures not already available in a specific language were translated and back-translated using standard procedures [34]. Final selection of instruments was agreed upon by consensus among the study team.

### Vaccination intention (Time 2 only)

At Time 2, participants were asked whether they would be willing to receive a COVID-19 vaccination once available to them. Responses were rated on a 7-point Likert scale from 1 (strongly disagree) to 7 (strongly agree).

### Socio-demographic variables (Time 1 only)

Assessed at Time 1, participants provided information on age (in years), gender, country of residence, employment status (working, not working), marital status and changes in financial situation during quarantine (better, same, or worse).

### Mental health diagnosis

Participants indicated whether they had received any mental health diagnosis since the start of the pandemic, including generalised anxiety, depression, panic disorder, obsessive-compulsive disorder, social anxiety, eating disorder, bipolar disorder, and/or other conditions.

### COVID-19 infection experience

Participants responded whether they, a partner, a family member, or a close contact had been infected with COVID-19.

### Perceived susceptibility and severity

Based on the Health Belief Model (HBM), perceived susceptibility (i.e., how much individuals believed they were susceptible to COVID-19) and perceived severity (i.e., how much an individual perceived COVID-19 to be a serious disease) were assessed using a modified version of an existing scale [35]. Each construct was measured with 3 items rated on a 5-point scale (1 = absolutely disagree, 5 = absolutely agree). Cronbach's alpha coefficient was 0.86, suggesting good internal consistency.

### COVID risk self-efficacy

Self-efficacy related to COVID-19 risk mitigation was measured using an adapted version of the New General Self-Efficacy Scale [36]. This 5-item scale used a 5-point Likert format (1 = strongly disagree to 5 = strongly agree). Cronbach's alpha coefficient was 0.88, suggesting good internal consistency.

### Perceived stress

Stress was measured using a 10-item Perceived Stress Scale [37], which assesses how unpredictable, uncontrollable, and overloaded participants found their lives during the past month. Responses were scored on a 5-point scale (1 = never,

5 = very often), with higher scores indicating greater perceived stress. Cronbach's alpha coefficient was 0.92, suggesting excellent internal consistency.

### Psychological flexibility

Psychological flexibility was measured using the PsyFlex scale [38], a 9-item instrument scored on a 5-point Likert scale. Items were reverse-coded, with higher scores indicating greater psychological flexibility. Cronbach's alpha coefficient was 0.91, suggesting excellent internal consistency.

### Positive and negative affect

Affect was assessed using the PANAS scale [39], expanded with five additional items (bored, confused, angry, frustrated, lonely) to capture pandemic-relevant affective states. Items were rated on a 7-point scale (1 = very little/not at all, 7 = extremely). Separate positive and negative affective scores were calculated. Cronbach's alpha coefficient was 0.95, suggesting excellent internal consistency.

### Pro-social behaviour

Prosociality was measured using six items from the Prosocialness Scale for Adults (PSA; [40]). Participants responded to statements regarding helping behaviours, empathy, and volunteering, rated on a 5-point scale. Higher scores reflected greater prosociality. Cronbach's alpha coefficient was 0.90, suggesting excellent internal consistency.

### Statistical analysis

We first examined whether vaccination intentions differed by sociodemographic characteristics, COVID-19 infection exposure (self, partner and significant other) and mental health diagnosis, using Kruskal Wallis run sum test. Changes in psychological variables between Time 1 (pre-rollout) and Time 2 (post-rollout) were assessed using paired t-tests. To examine predictors of vaccination intention, we fitted cumulative link models (CLMs), treating vaccination intention as an ordinal outcome. Models were estimated using the clm() function from the 'ordinal' package in R [41]. We specified four models to examine predictors of vaccination intentions. Model 1 included psychological predictors measured at Time 1 (before vaccination rollout), specifically self-efficacy, perceived severity, perceived susceptibility, perceived stress, psychological flexibility, prosociality, and positive and negative affect, along with sociodemographic covariates (age, gender, employment status, marital status, and financial changes) and mental health diagnoses. Model 2 used the same covariates but included the psychological predictors measured at Time 2 (after vaccination rollout). Model 3 was built upon Model 1 by additionally incorporating variables related to COVID-19 infection status (self, partner, significant other), while Model 4 extended Model 2 by including the same infection-related covariates. We examined intercorrelations among predictors to assess multicollinearity, and no problematic associations were found (r < .60; see Table 1) [42]. Psychological flexibility and prosociality were entered as independent predictors in all models. All analyses were conducted in R Statistical Software (v3.6.3; [43]).

## Results

### Sample characteristics

The final sample comprised 586 participants who met the study's inclusion criteria. Only participants who completed assessments at both time points were eligible for inclusion in the analyses. Participants with missing data on the primary outcome (vaccination intention) were excluded to ensure completeness and reliability of responses. The analytic sample was drawn from an initial dataset of N = 9,565 respondents, of whom 607 participants completed both pre- and post-vaccine rollout surveys. Following data screening, six people were removed from the dataset due to having more than 5%

**Table 1. Spearman correlations between the psycho-social factors before and after the Covid-19 vaccination rollout.**

| Variable | 1 | 2 | 3 | 4 | 5 | 6 | 7 | 8 | 9 | 10 | 11 | 12 | 13 | 14 | 15 |
|---|---|---|---|---|---|---|---|---|---|---|---|---|---|---|---|
| 1. Self-Efficacy-B | | | | | | | | | | | | | | | |
| 2. Self-Efficacy-FU | .52** | | | | | | | | | | | | | | |
| 3. Perceived Susceptibility-B | -.09 | -.1 | | | | | | | | | | | | | |
| 4. Perceived Susceptibility-FU | -.09 | -.1 | .58** | | | | | | | | | | | | |
| 5. Perceived Severity-B | -.05 | .03 | .54** | .44** | | | | | | | | | | | |
| 6. Perceived-Severity-FU | -.04 | .02 | .40** | .55** | .64** | | | | | | | | | | |
| 7. Perceived Stress-B | -.42** | -.32** | .13** | .16** | .09* | .13** | | | | | | | | | |
| 8. Perceived Stress-FU | -.34** | -.51** | .22** | .19** | .06 | .10* | .55** | | | | | | | | |
| 9. Positive Affect-B | .35** | .22** | -.03 | -.09 | -.07* | -.11** | -.45** | -.33** | | | | | | | |
| 10. Positive Affect-FU | .29** | .31** | -.15** | -.10* | -.12** | -.15** | -.31** | -.50** | .64** | | | | | | |
| 11. Negative Affect-B | -.34** | -.31** | .24** | .23** | .15** | .17** | .64** | .48** | -.35** | -.27** | | | | | |
| 12. Negative Affect-FU | -.32** | -.40** | .25** | .23** | .10* | .15** | .46** | .71** | -.29** | -.40** | .63** | | | | |
| 13. Prosociality-B | .19** | .16** | -.03 | -.05 | -.06 | -.11** | -.06 | -.04 | .25** | .21** | -.08* | -.04 | | | |
| 14. Prosociality-FU | .16** | .19** | -.06 | -.08* | -.11** | -.14** | -.05 | -.10** | .21** | .28** | -.12** | -.11** | .76** | | |
| 15. Psychological Flexibility-B | .41** | .34** | -.16** | -.16** | -.10* | -.15** | -.53** | -.44** | .58** | .45** | -.50** | -.38** | .27** | .23** | |
| 16. Psychological Flexibility-FU | .34** | .44** | -.18** | -.14** | -.04 | -.09 | -.43** | -.63** | .46** | .60** | -.42** | -.55** | .20** | .26** | .68** |

* Indicates p < .05; ** indicates p < .01, Baseline (B), Follow-up (FU).

of missing data [44], and additionally, 15 participants were removed as statistical outliers. This resulted in a final sample of $N = 586$ individuals (16.9% male), aged between 18 – 79 ($M = 39.32$, $SD = 13.43$). There were no significant differences in the vaccination intentions across categories of gender, financial situation, employment status, marital status, personal COVID-19 infection history, partner or significant other's infection status, or self-reported mental health diagnosis (see Table 2 for full results).

## Changes in psycho-social factors following COVID-19 vaccination rollout

Paired sample t-tests were used to examine changes in psycho-social variables between the two assessment points. Participants reported significantly lower self-efficacy for following recommended protective behaviours after the vaccine rollout compared to the initial phase of the pandemic ($t$ (595) =4.65, $p < .001$). There were also significant decreases in prosociality ($t$ (595) =6.79, $p < .001$), psychological flexibility ($t$ (595) =2.84 $p < .01$) and positive affect ($t$(595)=2.46, $p < .05$). No other psychosocial variables showed statistically significant differences between Time 1 and Time 2 (see Table 3).

## Predictors of COVID-19 vaccination intentions

The cumulative link model indicated that self-efficacy to follow recommended protective behaviours was positively associated with vaccination intentions both before the vaccination rollout (OR = 1.31, 95% CI [1.02, 1.70], $p < .05$) and after (OR = 1.36, 95% CI [1.06, 1.75], $p < .05$), suggesting that higher self-efficacy was associated with about 30–36% greater odds of vaccination intention. Individuals with higher confidence in their ability to maintain social distancing and stay home were more likely to intend to vaccinate. Perceived severity of COVID-19 also showed a significant positive association with vaccination intentions before (OR = 1.18, 95% CI [1.11, 1.25], $p < .001$) and after (OR = 1.21, 95% CI [1.14, 1.28], $p < .001$) the rollout, reflecting a consistent moderate effect. Participants who viewed COVID-19 as more severe were consistently more inclined to vaccinate. In contrast, perceived susceptibility to infection was negatively associated with vaccination intentions before the rollout (OR = 0.91, 95% CI [0.86, 0.97], $p < .001$), suggesting that those who felt at higher risk were

**Table 2. Differences in sociodemographic characteristics and covid-19 vaccination intentions.**

| | Overall (N = 586) | Vaccination Intent | | | | | | | Kruskal Wallis (χ²) | p-value |
|---|---|---|---|---|---|---|---|---|---|---|
| | | 1 (%) | 2(%) | 3(%) | 4(%) | 5(%) | 6(%) | 7(%) | | |
| **Gender** | | | | | | | | | | |
| Males | 99(16.9) | 7(1.2) | 4 (.67) | 2(.33) | 6(1.0) | 7(1.2) | 9(1.5) | 64(11.0) | .09 | .76 |
| Females | 487(83.1) | 31(5.2) | 16(3.0) | 10(1.85) | 46(7.8) | 36(6.1) | 42(7.1) | 306(52.4) | | |
| **Finances** | | | | | | | | | | |
| Better | 110(18) | 3(.5) | 3(.50) | 3(.50) | 8(1.4) | 6(1.0) | 12(2.0) | 75(12.8) | 2.83 | .23 |
| Same | 369(63) | 27(4.6) | 7(1.2) | 8(1.4) | 38(6.5) | 28(4.87) | 30(5.1) | 231(39.4) | | |
| Worse | 107(19) | 8(1.4) | 10(1.7) | 1(.2) | 6(1.0) | 9(1.5) | 9(1.5) | 64(10.9) | | |
| **Employment** | | | | | | | | | | |
| Working | 358(61) | 25(4.3) | 15(2.6) | 6(1.0) | 36(6.1) | 25(4.3) | 26(4.4) | 225(38.4) | .40 | .53 |
| Not Working | 228(39) | 13(2.2) | 5(.9) | 6(1.0) | 16(2.7) | 18(3.1) | 25(4.3) | 145(24.7) | | |
| **Marital Status** | | | | | | | | | | |
| Single | 219(37) | 16(2.7) | 6(1.0) | 5(0.9) | 24(4.1) | 18(3.1) | 23(3.9) | 127(21.7) | 3.19 | .07 |
| Couple | 367(63) | 22(3.8) | 14(2.4) | 7(1.2) | 28(4.8) | 25(4.3) | 28(4.8) | 243(41.5) | | |
| **Covid infection-Self** | | | | | | | | | | |
| Yes | 7(1.2) | 1(.17) | 0 | 0 | 0 | 0 | 2(.34) | 4(.68) | .11 | .94 |
| No | 508(86.7) | 33(5.6) | 18(3.1) | 11(1.88) | 46(7.85) | 33(5.6) | 44(7.5) | 323(55.11) | | |
| Not sure | 71(12.1) | 4(.68) | 2(.34) | 1(.17) | 6(1.0) | 10(1.7) | 5(.85) | 43(7.3) | | |
| **Covid infection-Partner** | | | | | | | | | | |
| Yes | 8(1) | 3(.5) | 0 | 0 | 0 | 0 | 1(.2) | 4(.7) | 1.49 | .47 |
| No | 537(92) | 34(5.5) | 18(3.0) | 11(2.02) | 48(8.1) | 39(6.6) | 47(7.9) | 340(58.5) | | |
| Not sure | 39(7) | 1(.2) | 2(.3) | 0 | 4(.7) | 4(.7) | 3(.5) | 25(4.3) | | |
| **Covid infection-Significant Other** | | | | | | | | | | |
| Yes | 40(7) | 3(.51) | 4(.68) | 0 | 3(.51) | 2(.34) | 3(.51) | 25(4.27) | .35 | .83 |
| No | 485(83) | 32(5.46) | 15(2.56) | 11(1.88) | 47(8.02) | 32(5.46) | 42(7.16) | 306(52.22) | | |
| Not sure | 61(10) | 3(.51) | 1(.17) | 1(.17) | 2(.34) | 9(1.53) | 6(1.02) | 39(6.65) | | |
| **Diagnosed Mental Disorders** | | | | | | | | | | |
| Yes | 121(20.6) | 9(1.5) | 6(1.0) | 1(.2) | 12(2.0) | 12(2.0) | 7(1.2) | 74(12.6) | | |
| Anxiety Disorder(s) | 45 (7.5) | 3(.5) | 1(.2) | 0 | 6(1.0) | 4(.7) | 2(.3) | 28(4.8) | .53 | .46 |
| Depression | 13 (2.3) | 1(.2) | 1(.2) | 1(.2) | 0 | 0 | 2(.3) | 8(1.4) | | |
| Anxiety/Depression | 15(2.3) | 0 | 0 | 0 | 2(.3) | 2(.3) | 1(.2) | 9(1.5) | | |
| Other disorders (ED, OCD, BD) | 35(5.9) | 3(.5) | 3(.5) | 0 | 4(.7) | 2(.3) | 2(.3) | 21(3.6) | | |
| Comorbidities | 15 (2.6) | 2(.3) | 1(.2) | 0 | 0 | 4(.7) | 0 | 8(1.4) | | |
| No | 465 (79.3) | 29(4.9) | 14(2.4) | 11(1.9) | 40(6.8) | 31(5.3) | 44(7.5) | 296(50.5) | | |

*Note.* Vaccination intention rated on a 7-point scale (1 = very unlikely to 7 = very likely to vaccinate). Group differences tested using Kruskal–Wallis rank-sum tests.

less likely to intend to vaccinate by about 9%. This relationship was no longer significant after the rollout (OR = 0.96, 95% CI [0.90, 1.02], p = .164). No other psychological factors measured before or after the vaccination rollout were significantly associated with vaccination intentions (Table 4).

## Discussion

This study examined how psychosocial and social cognitive factors influenced COVID-19 vaccination intentions at two time points, before and after the vaccination rollout. We observed declines in self-efficacy, prosociality, psychological

**Table 3. Changes in psycho-social factors before and after the COVID-19 vaccination rollout.**

| | Before (M±SD) | After (M±SD) | t-test (595) | P-value | Cohen's d |
|---|---|---|---|---|---|
| COVID-19 related Self-Efficacy | 6.19±0.77 | 6.04±0.82 | **4.65** | <.001 | .19 |
| Perceived severity | 12.44±3.63 | 12.41±3.76 | .26 | .80 | .01 |
| Perceived susceptibility | 8.70±3.45 | 8.66±3.50 | .34 | .73 | .01 |
| Perceived stress | 16.47±7.53 | 16.54±7.60 | -.24 | .81 | <.01 |
| Pro-sociality | 22.71±4.20 | 21.90±4.24 | **6.79** | <.001 | .28 |
| Psychological Flexibility | 33.52±5.54 | 33.02±5.77 | **2.84** | <.01 | .11 |
| Negative Affect | 29.33±10.82± | 29.58±11.36 | -.65 | .51 | -.03 |
| Positive Affect | 28.76±8.03 | 28.10±8.41 | 2.46 | .014 | .10 |

*M* (Mean), *SD* (Standard Deviation).

flexibility, and positive affect following the rollout. Despite these declines, self-efficacy and perceived severity consistently predicted vaccination intentions across both time points. In contrast, perceived susceptibility predicted lower vaccination intention only before the rollout. No other psychological factors were significant predictors.

## Shifts in emotional and psychosocial factors across the pandemic

Participants' self-efficacy, prosociality, psychological flexibility and positive affect were reduced after vaccination rollout compared to the early phase of the pandemic. These findings align with growing evidence of pandemic-related psychological fatigue and behavioural adaptation, which describe how sustained stress, uncertainty and prolonged restrictions can reduce motivation and perceived behavioural control over time [45–48]. Previous studies have reported increased emotional exhaustion and reduced well-being due to sustained uncertainty, social restrictions and inconsistent policy responses [49–55]. The extended duration of the crisis – combined with evolving public health messages, policy inconsistencies, and social disruption- likely contributed to the reduction of motivation, adaptability and positive emotional states [56–58].

Importantly, the decline in self-efficacy to follow recommended protective behaviours may also signal a shift in public focus – from collective preventive efforts (e.g., mask-wearing, distancing), to personal protection through vaccination. This change may also be linked to a sense of complacency that emerged following vaccination availability, as suggested in previous studies [59,60]. Consequently, this shift in mindset may have contributed to the reduced engagement in prosocial behaviour and flexible coping strategies [61]. Our findings support this interpretation, while self-efficacy remained positively associated with vaccination intentions, indicators of collective motivation declined after vaccination rollout.

## Predictors of vaccination intentions

Despite the decline in various psychosocial factors, self-efficacy remained a robust and consistent predictor of vaccination. Individuals who felt more capable of adhering to recommended protective behaviours were more likely to express willingness to vaccinate, both before and after the availability of vaccines. This finding supports prior research demonstrating that self-efficacy is a key determinant of vaccine-related decision-making [11,22,62] and extends its relevance across two temporally distinct phases of a public health crisis. For example, a cross-sectional study in the US during early 2021 similarly found that self-efficacy was positively associated with vaccination uptake and intention [63]. These results emphasise the importance of public health strategies that enhance individuals' confidence in their ability to engage in protective behaviours – a central component of behaviour change and vaccination promotion.

However, our findings contrast with a longitudinal study conducted in New Zealand between February 2021 and May 2021, which reported that self-efficacy was negatively associated with COVID-19 vaccine hesitancy at both times [6]. One

Table 4. Cumulative link model results on the effects of psychosocial factors, sociodemographic and COVID-19 infection variables on vaccination intentions before and after the rollout.

| | Vaccination Intentions | | | | | | | |
| | Model 1 | | Model 2 | | Model 3 | | Model 4 | |
| | OR 95% CI | p-value | OR 95% CI | p-value | OR 95% CI | p-value | OR 95% CI | p-value |
|---|---|---|---|---|---|---|---|---|
| **Pre-Vaccination rollout** | | | | | | | | |
| Covid Self-Efficacy | 1.30 [1.01 – 1.67] | **0.041** | | | 1.31 [1.02 – 1.70] | **0.036** | | |
| Perceived severity | 1.18 [1.11 – 1.25] | **<0.001** | | | 1.18 [1.11 – 1.25] | **<0.001** | | |
| Perceived susceptibility | 0.91 [0.86 – 0.97] | **0.004** | | | 0.91 [0.86 – 0.97] | **0.005** | | |
| Perceived stress | 1.01 [0.98 – 1.04] | 0.564 | | | 1.01 [0.98 – 1.05] | 0.548 | | |
| Pro-sociality | 1.01 [0.97 – 1.05] | 0.729 | | | 1.01 [0.96 – 1.05] | 0.770 | | |
| Psychological Flexibility | 0.96 [0.92 – 1.00] | 0.062 | | | 0.96 [0.92 – 1.00] | 0.062 | | |
| Negative Affect | 1.00 [0.98 – 1.02] | 0.695 | | | 1.00 [0.97 – 1.02] | 0.644 | | |
| Positive Affect | 1.01 [0.98 – 1.04] | 0.416 | | | 1.01 [0.98 – 1.04] | 0.397 | | |
| Age | 1.01 [0.99 – 1.02] | 0.387 | 1.00 [0.99 – 1.02] | 0.555 | 1.01 [0.99 – 1.02] | 0.416 | 1.00 [0.99 – 1.02] | 0.594 |
| Gender [Female] | 0.93 [0.58 – 1.48] | 0.755 | 0.77 [0.48 – 1.23] | 0.268 | 0.96 [0.60 – 1.53] | 0.851 | 0.80 [0.50 – 1.28] | 0.350 |
| Finances [Same] | 0.89 [0.56 – 1.41] | 0.614 | 0.92 [0.58 – 1.47] | 0.732 | 0.90 [0.56 – 1.43] | 0.652 | 0.94 [0.59 – 1.51] | 0.800 |
| Finances [Worse] | 0.68 [0.39 – 1.21] | 0.191 | 0.66 [0.37 – 1.19] | 0.167 | 0.69 [0.39 – 1.22] | 0.201 | 0.67 [0.37 – 1.20] | 0.173 |
| Employment Status [Yes] | 0.83 [0.58 – 1.19] | 0.313 | 0.90 [0.62 – 1.28] | 0.547 | 0.84 [0.59 – 1.21] | 0.361 | 0.91 [0.63 – 1.30] | 0.601 |
| Marital Status [Single] | 0.83 [0.58 – 1.18] | 0.298 | 0.89 [0.63 – 1.27] | 0.535 | 0.82 [0.58 – 1.17] | 0.281 | 0.91 [0.63 – 1.30] | 0.602 |
| Mental Disorders [Yes] | 0.79 [0.51 – 1.21] | 0.278 | 0.87 [0.56 – 1.34] | 0.530 | 0.79 [0.52 – 1.22] | 0.293 | 0.88 [0.57 – 1.36] | 0.561 |
| Covid - self [No] | | | | | 0.41 0.04 – 3.77] | 0.434 | 0.51 [0.06 – 4.56] | 0.546 |
| Covid - self [Notsure] | | | | | 0.42 0.04 – 4.13] | 0.458 | 0.38 [0.04 – 3.69] | 0.407 |
| Covid - partner [No] | | | | | 5.54 [0.70 – 44.00] | 0.105 | 5.52 [0.75 – 40.70] | 0.094 |
| Covid - partner [Notsure] | | | | | 5.34 [0.55 – 51.88] | 0.149 | 7.23 [0.82 – 63.95] | 0.075 |
| Covid - significant other [No] | | | | | 1.00 [0.48 – 2.08] | 0.999 | 0.95 [0.45 – 1.98] | 0.886 |
| Covid – significant other [Notsure] | | | | | 1.04 [0.41 – 2.65] | 0.934 | 0.94 [0.37 – 2.39] | 0.903 |
| **Post-Vaccination rollout** | | | | | | | | |
| Covid Self-Efficacy | | | 1.36 [1.07 – 1.74] | **0.013** | | | 1.36 [1.06 – 1.75] | **0.015** |
| Perceived severity | | | 1.20 [1.13 – 1.27] | **<0.001** | | | 1.21 [1.14 – 1.28] | **<0.001** |
| Perceived susceptibility | | | 0.96 [0.91 – 1.02] | 0.211 | | | 0.96 [0.90 – 1.02] | 0.164 |
| Perceived stress | | | 1.02 [0.98 – 1.06] | 0.372 | | | 1.02 [0.98 – 1.06] | 0.399 |
| Pro-sociality | | | 1.01 [0.97 – 1.06] | 0.506 | | | 1.01 [0.97 – 1.06] | 0.582 |
| Psychological Flexibility | | | 0.99 [0.95 – 1.04] | 0.679 | | | 0.99 [0.94 – 1.03] | 0.593 |
| Negative Affect | | | 1.00 [0.98 – 1.02] | 0.874 | | | 1.00 [0.98 – 1.02] | 0.895 |
| Positive Affect | | | 1.00 [0.97 – 1.03] | 0.958 | | | 1.00 [0.98 – 1.03] | 0.860 |

*Note*: OR = odds ratio; CI = confidence interval. Brackets indicate baseline reference categories. Model 1 = Pre-rollout psychosocial predictors; Model 2 = Post-rollout psychosocial + sociodemographic predictors; Model 3 = Model 1 + COVID-19 infection variables; Model 4 = Model 2 + COVID-19 infection variables.

(Continued)

possible explanation for this discrepancy lies in the conceptualisation of self-efficacy. While some studies focus specifically on vaccine-related self-efficacy, our study assessed general self-efficacy for following pandemic-related protective behaviours. This broader conceptualisation may capture a wider behavioural orientation where individuals confident in their ability to engage in protective measures in general, are also more likely to get trust and engage in vaccination programmes.

We also found that participants who perceived COVID-19 as more severe were more likely to take the vaccine, both before and after rollout. This finding is consistent with existing literature demonstrating the role of perceived severity in motivating health-protective behaviours [23,64,65]. It is likely that individuals' health decisions are influenced by how seriously they perceive a health threat's consequences [64]. This interpretation aligns with the HBM [15], suggesting that individuals are more likely to take preventive actions, such as vaccination, when they perceive a health threat as severe. This perception of severity can be a powerful motivator for vaccination and can inform public health efforts to combat the spread of future viruses.

Interestingly, contrary to our hypotheses, individuals who perceived themselves as being at higher risk of contracting COVID-19 were less likely to intend to vaccinate before the rollout. One possible explanation is that heightened perceived susceptibility may have been linked to greater anxiety, mistrust, or concerns about vaccine safety when vaccines were not yet available. Once vaccinations became accessible and more information about their safety and efficacy was disseminated, this negative association disappeared, suggesting that availability and public communication may have mitigated initial hesitancy among high-risk individuals. Therefore, individuals with high perceived susceptibility were more cautious or uncertain about vaccination safety, especially in the early phase before long-term data on side effects were available [66]. These findings underscore the importance of distinguishing between disease-related and vaccine-related risk in public communication. During early phases of vaccine development, individuals who perceived themselves as highly susceptible to COVID-19 may have also experienced heightened anxiety and uncertainty about vaccine safety, amplifying hesitation. Tailored communication strategies that acknowledge such fears, transparently address safety data, and emphasise the protective benefits of vaccination could help prevent this paradoxical effect in future health crises.

Typically, perceived susceptibility increases motivation for preventive action by heightening personal relevance and risk salience. However, during the pre-rollout phase, heightened susceptibility may have amplified anxiety and uncertainty in the absence of clear safety information, particularly amid mixed messages and misinformation about vaccine development. As vaccines became available and credible data on safety and efficacy were disseminated, this negative association diminished, suggesting that transparent, trust-building communication can restore the usual motivational role of risk perception.

A similar pattern was observed in a Swiss study conducted between March and April 2020, which reported a negative association between risk perception and protective intentions [67]. The authors suggested that during lockdown, individuals' public and private lives were affected, thus driving risk perceptions to lose their relevance for intention formation and behaviour. The same study found that self-efficacy and response efficacy (i.e., individuals' expectation that a protective behaviour will effectively reduce the risk) were the most important predictors for intentions and protective behaviours.

It's plausible that individuals who experienced COVID-19, or had significant others who did, may have experienced the illness in a manner different from their expectations. This could have influenced their perception of the disease's susceptibility and, consequently, their motivation for vaccination. Over time, this might have contributed to a diminished urgency or willingness to get vaccinated. However, in our study (see Table 2), the direct relationship between personal COVID-19 experiences and vaccination intentions was not evident. Future studies should consider examining how personal experiences with COVID-19 influence vaccination intentions, and how other psychological or social mediators may impact vaccination intentions.

Although several psychosocial factors such as perceived stress, prosociality, psychological flexibility, and affect decreased after the vaccination rollout, these variables did not significantly predict vaccination intentions at either time point. This suggests that while these constructs capture broader emotional adaptation to the pandemic [68,69], their

influence on specific behavioural intentions may diminish once vaccines become available. As the focus of public discourse shifted from collective coping and emotional resilience to concrete health decision-making, more targeted cognitive appraisals, such as perceived severity and self-efficacy, may have become stronger proximal determinants of vaccination behaviour [70]. This pattern highlights how general psychological well-being and motivation can fluctuate independently from specific health-related intentions.

Furthermore, the limited predictive power of these broader psychosocial factors may also reflect statistical competition with more proximal, vaccine-specific cognitions. When general psychological states, such as stress, affect, or psychological flexibility, are considered alongside direct vaccine-related predictors (e.g., perceived safety, confidence, collective responsibility, and trust in authorities), much of their variance is absorbed by these more specific constructs. Confidence in vaccine safety and perceived collective responsibility have been identified as dominant predictors of vaccine uptake, overshadowing broader emotional dispositions, while belief-based and informational factors, rather than general psychological traits, best explain patterns of hesitancy [71,72]. Together, these findings suggest that general psychological factors may shape vaccination intentions indirectly, primarily through their influence on targeted cognitive appraisals and vaccine-specific beliefs.

Given the multinational nature of this study, it is important to consider that psychosocial predictors of vaccination intentions may vary across cultural and policy contexts. Societies characterised by stronger collectivist orientations or higher institutional trust often display greater prosocial motivation and adherence to public health recommendations, whereas more individualistic or distrustful environments may amplify hesitancy despite similar risk perceptions [73,74]. The declines in self-efficacy and prosociality found here could therefore reflect differing national experiences of pandemic management and collective fatigue. Recent population-level evidence from the United States likewise shows that mental and physical health recovery during the pandemic followed non-linear, uneven trajectories, indicating broader patterns of psychosocial disruption and adaptation [32]. Future research should employ multilevel or cross-cultural modelling to examine how national-level factors, such as policy stringency, vaccination availability, and trust in authorities, interact with individual level beliefs and emotions to shape vaccination behaviour.

## Strengths and limitations

This is, to our knowledge, the first multinational study to examine both social-cognitive and emotional predictors of COVID-19 vaccination intention before and after vaccination rollout. Among the study's key strengths are its diverse, cross-national sample and its prospective design, allowing for comparisons before and after vaccination rollout. However, several limitations must be noted. First, although the sample spanned 33 countries, we did not control for country-level differences such as policy, vaccination access, or cultural norms. Future studies should incorporate multilevel modelling or stratified analyses to assess cross-country variation more systematically. Second, the sample was predominantly female, which may limit generalisability. Emerging evidence indicates that women have shown greater vaccine hesitancy and lower uptake than men across several contexts, often linked to heightened safety concerns and lower institutional trust [75,76]. Consequently, the predominance of women in our sample may have led to slightly lower overall vaccination intention levels, potentially underestimating intentions in more gender-balanced populations. Future studies should therefore aim for more balanced recruitment and investigate potential gender-based differences in psychosocial predictors of vaccination behaviour. Third, all data were self-reported and subject to bias, including the social desirability effect. Fourth, the online, opportunistic recruitment strategy may have underrepresented individuals who are less digitally engaged or more vaccine-hesitant. Finally, vaccination intention was assessed only at the second time, and actual vaccination uptake was not monitored.

## Implications

This study's findings highlight the importance of self-efficacy in communicating strategies related to COVID-19 vaccinations as well as future vaccinations [77,78]. Based on the literature, there are two ways to improve self-efficacy [63]. First, by targeting individuals' beliefs directly through interventions that help with decision-making and secondly, by modifying

the external environment, such as the circumstances, people, things, and events around them that influence their decisions. Educational programs, public awareness campaigns, and modelling interventions such as demonstrating vaccination benefits or important others who vaccinate, can potentially improve people's beliefs [79,80], but these could be grounded on shifting people's perspectives (such as the severity of a health threat) rather than just providing information. Additionally, assigning informed stakeholders to key roles within organisations and governmental positions can provide support to policies promoted by governments, especially those aimed at overcoming vaccination barriers [63]. Campaigns could include useful information about both the health threat (e.g., COVID-19) and the vaccine, emphasising vaccination safety. These promotional efforts could facilitate higher vaccination uptake rates against future pandemics or epidemics by targeting changes in individuals' attitudes and self-efficacy. In addition to public awareness campaigns, disseminating targeted information about COVID-19 vaccination safety could also be facilitated through health education programs targeting young people to help them with decision-making. Collectively, these interventions could mitigate the adverse attitudes related to the COVID-19 vaccination or any new vaccine.

## Acknowledgments

We would like to thank all the individuals who participated in the study.

## Author contributions

**Conceptualization:** Maria Karekla, Andrew Gloster, Angelos Kassianos.

**Data curation:** Maria Kyprianidou, Demetris Lamnisos.

**Formal analysis:** Athina Manoli, Maria Kyprianidou.

**Investigation:** Jelena Lubenko, Giovambattista Presti, Valeria Squatrito, Marios Constantinou, Christiana Nicolaou, Savvas Papacostas, Gökçen Aydın, Yuen Yu Chong, Wai Tong Chien, Ho Yu Cheng, Francisco Ruiz, Maria Belen Garcia-Martin, Diana Obando-Posada, Miguel Segura-Vargas, Vasilis S. Vasiliou, Louise McHugh, Stefan Höfer, Adriana Băban, David Dias Neto, Ana Nunes Da Silva, Jean-Louis Monestès, Javier Alvarez-Galvez, Marisa Paez-Blarrina, Francisco Montesinos, Sonsoles Valdivia-Salas, Dorottya Ori, Bartosz Kleszcz, Raimo Lappalainen, Iva Ivanović, David Gosar, Frederick Dionne, Rhonda Merwin, Maria Karekla, Andrew Gloster, Angelos Kassianos.

**Methodology:** Maria Kyprianidou, Maria Karekla, Andrew Gloster, Angelos Kassianos.

**Project administration:** Maria Karekla, Andrew Gloster, Angelos Kassianos.

**Supervision:** Angelos Kassianos.

**Writing – original draft:** Athina Manoli, Maria Kyprianidou, Angelos Kassianos.

**Writing – review & editing:** Athina Manoli, Maria Kyprianidou, Demetris Lamnisos, Jelena Lubenko, Giovambattista Presti, Valeria Squatrito, Marios Constantinou, Christiana Nicolaou, Savvas Papacostas, Gökçen Aydın, Yuen Yu Chong, Wai Tong Chien, Ho Yu Cheng, Francisco Ruiz, Maria Belen Garcia-Martin, Diana Obando-Posada, Miguel Segura-Vargas, Vasilis S. Vasiliou, Louise McHugh, Stefan Höfer, Adriana Băban, David Dias Neto, Ana Nunes Da Silva, Jean-Louis Monestès, Javier Alvarez-Galvez, Marisa Paez-Blarrina, Francisco Montesinos, Sonsoles Valdivia-Salas, Dorottya Ori, Bartosz Kleszcz, Raimo Lappalainen, Iva Ivanović, David Gosar, Frederick Dionne, Rhonda Merwin, Maria Karekla, Andrew Gloster, Angelos Kassianos.

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
