## [Decision Letter · Decision Letter 0]

29 Oct 2025

PGPH-D-25-02717

Beyond Emotions: Social Cognitive Predictors of COVID-19 Vaccination Intentions Before and After Vaccine Roll-Out

Dear Dr. Manoli,

Thank you for submitting your manuscript to PLOS Global Public Health. After careful consideration, we feel that it has merit but does not fully meet PLOS Global Public Health’s publication criteria as it currently stands. Therefore, we invite you to submit a revised version of the manuscript that addresses the points raised during the review process.

The authors raised some similar points about your article. If you feel any reviewer comments are in conflict, or if you do not want to respond to a certain point, just raise that in your response. You also do not need to cite a particular article that a reviewer gave, although you can if you find it useful.

We look forward to receiving your revised manuscript.

Kind regards,

Abram L. Wagner, PhD, MPH

Academic Editor

Journal Requirements:

1. We ask that a manuscript source file is provided at Revision. Please upload your manuscript file as a .doc, .docx, .rtf or .tex.

2. In the online submission form, you indicated that “The data contain potentially identifying or sensitive participant information and cannot be shared publicly. Data are available from the corresponding author for researchers who meet the criteria for access to confidential data.”.

a. In a public repository,

b. Within the manuscript itself, or

c. Uploaded as supplementary information.

Additional Editor Comments (if provided):

Reviewers' comments:

Reviewer's Responses to Questions

**Comments to the Author**

1. Does this manuscript meet PLOS Global Public Health’s publication criteria?

Reviewer #1: Yes

Reviewer #2: Yes

Reviewer #3: Yes

2. Has the statistical analysis been performed appropriately and rigorously?

Reviewer #1: Yes

Reviewer #2: Yes

Reviewer #3: Yes

3. Have the authors made all data underlying the findings in their manuscript fully available (please refer to the Data Availability Statement at the start of the manuscript PDF file)?

Reviewer #1: Yes

Reviewer #2: No

Reviewer #3: No

4. Is the manuscript presented in an intelligible fashion and written in standard English?

Reviewer #1: Yes

Reviewer #2: Yes

Reviewer #3: Yes

Reviewer #1: No comments. The paper is well written and all the different areas including introduction, materials and methods, results, data analysis and discussions arer well written and well articulated. The paper covers all areas well and succintcly.

Reviewer #2: 1. The intro section could be tightened slightly by reducing redundancy in references describing early-pandemic hesitancy trends and focusing on what remains unknown or understudied..specifically, within-person change over time, recommend adding these new studies which cover this perspective

2. Strengthen the discussion to address possible cross-cultural differences in psychosocial predictors, given the study’s multinational scope

Additionally, recommend citing https://doi.org/10.1136/bmjph-2025-002765, new 2025 study which provides complementary evidence on population-level disruption and recovery during the pandemic, aligning well with the observed declines in self-efficacy and pro sociality reported in this manuscript

3. The results section could be improved by adding a short statement on effect sizes or confidence intervals for the main predictors, enhancing interpretability and policy relevance

4. Consider clarifying whether psychological flexibility and pro sociality were treated as independent or interrelated constructs in the models, as their conceptual overlap could influence mediation pathways

5. A brief mention of pandemic fatigue or behavioral adaptation literature (e.g., sustained stress leading to reduced self-efficacy) could provide a more nuanced interpretation of the post-rollout decline in psychosocial measures

6. Regarding the data availability statement, it currently notes that access will be provided to “researchers who meet the eligibility criteria.” This phrasing is vague and does not fully align with the journal’s open-data policy. The authors should clarify the specific access conditions, including who determines eligibility and how qualified researchers can request the dataset

7. Tables 2–4 would benefit from clearer labeling of model variables and consistent reporting of p-values and ORs to improve readability

Reviewer #3: The manuscript, titled 'Beyond Emotions: Social Cognitive Predictors of COVID-19 Vaccination Intentions Before and After Vaccine Roll-Out,' appears to meet the publication criteria by describing methodologically rigorous research. The study utilised a multinational prospective design across 33 countries, examining social-cognitive and emotional predictors of COVID-19 vaccination intentions before and after vaccine rollout. This approach allows for comparisons across two distinct phases of the pandemic, enhancing the robustness of the findings. The conclusions drawn, particularly regarding the consistent predictive power of self-efficacy and perceived severity, are directly supported by the data presented. The study also highlights declines in self-efficacy, pro-sociality, psychological flexibility, and positive affect after vaccination rollout, which aligns with evidence of pandemic-related psychological fatigue.

The study was approved by the Cyprus National Bioethics Committee (Π 2020.01.60), and all participants provided informed consent electronically before participation, indicating ethical conduct.

The statistical analysis appears to be performed appropriately and rigorously. Changes in psychological variables between the pre-rollout (Time 1) and post-rollout (Time 2) phases were assessed using paired t-tests. To predict vaccination intention, cumulative link models (CLMs) were fitted, treating vaccination intention as an ordinal outcome, and estimated using the 'ordinal' package in R. Four models were specified, progressively incorporating psychological predictors, sociodemographic covariates, and COVID-19 infection status variables. Intercorrelations among predictors were examined to assess multicollinearity, with no problematic associations found (r < .60). This detailed approach suggests a comprehensive and rigorous statistical methodology.

The manuscript explicitly states that the data contain potentially identifying or sensitive participant information and cannot be shared publicly. Instead, data are available from the corresponding author for researchers who meet the criteria for access to confidential data. This aligns with PLOS policy for sensitive data, where full public availability might be restricted due to ethical concerns. The authors have provided a transparent statement regarding data access.

The manuscript is presented in an intelligible fashion and generally written in standard English. The language is clear, and the structure is logical, making the content easy to follow. No significant typographical or grammatical errors were noted that would impede understanding.

The study's multinational, prospective design is a significant strength, allowing for a robust examination of vaccination intentions across different phases of the pandemic and diverse populations. The identification of self-efficacy and perceived severity as consistent predictors of vaccination intention, even as other psychosocial factors declined, provides valuable insights for public health messaging.

Suggestions for Improvement:

Generalisability of Sample: While the study includes a diverse, cross-national sample, the authors acknowledge that the sample was predominantly female, which may limit generalizability. Further discussion of how this demographic imbalance might influence the findings, or how future research could address it, would be beneficial.

Control for Country-Level Differences: The authors note that country-level differences (e.g., policy, vaccination access, cultural norms) were not controlled for. While this is a limitation of the current study, it would be helpful to briefly elaborate on the potential implications of these uncontrolled variables on the observed results and reinforce the importance of multilevel modelling or stratified analyses in future research.

Perceived Susceptibility: The finding that perceived susceptibility was negatively associated with vaccination intentions before rollout, but not after, is intriguing. The explanation provided (anxiety, mistrust, or concerns about vaccine safety when vaccines were not yet available) is plausible. Expanding on this point, perhaps with a brief discussion of how public health communication could specifically address these early concerns about susceptibility and safety, could add practical value.

Psychological Variables as Non-Significant Predictors: The manuscript states that other psychological variables (e.g., perceived stress, psychological flexibility, pro-sociality, positive and negative affect) were not significant predictors of intentions. Given that some of these factors declined post-rollout, a more in-depth discussion on why they did not predict intention, despite their shifts, could provide further nuance to the findings.

Finally, the manuscript presents methodologically sound and ethically rigorous research, with conclusions well-supported by the data. The statistical analysis is appropriate, and the data availability statement is clear. The paper is well-written, and the suggested improvements aim to further enhance the depth and practical implications of the findings.

**Do you want your identity to be public for this peer review?** For information about this choice, including consent withdrawal, please see our Privacy Policy

Reviewer #1: **Yes: ** Samoel Ashimosi Khamadi

Reviewer #2: No

Reviewer #3: **Yes: ** Ishrat Islam

---

## [Editor Report · Decision Letter 1]

3 Dec 2025

Beyond Emotions: Social Cognitive Predictors of COVID-19 Vaccination Intentions Before and After Vaccine Roll-Out

PGPH-D-25-02717R1

Dear Dr Manoli,

We are pleased to inform you that your manuscript 'Beyond Emotions: Social Cognitive Predictors of COVID-19 Vaccination Intentions Before and After Vaccine Roll-Out' has been provisionally accepted for publication in PLOS Global Public Health.

Best regards,

Abram L. Wagner, PhD, MPH

Academic Editor